# Malignant Post-Transplant Lymphoproliferative Disorder of Nasopharynx in Myelodysplastic Disorder

**DOI:** 10.3390/healthcare9020217

**Published:** 2021-02-17

**Authors:** Chih-Wei Luan, Chih-Cheng Chen, Kam-Fai Lee, Ming-Shao Tsai, Yao-Te Tsai, Cheng-Ming Hsu, Geng-He Chang

**Affiliations:** 1Department of Otorhinolaryngology Head and Neck Surgery, Lo Sheng Sanatorium and Hospital Ministry of Health and Welfare, New Taipei City 242, Taiwan; jackluan2010@gmail.com; 2Graduate Institute of Clinical Medical Sciences, College of Medicine, Chang Gung University, Taoyuan 333, Taiwan; b87401061@cgmh.org.tw; 3Division of Hematology and Oncology, Chang Gung Memorial Hospital, Chiayi Branch 613, Taiwan; ccchen@cgmh.org.tw; 4Department of Pathology, Chang Gung Memorial Hospital, Chiayi Branch 613, Taiwan; Ikf2002@cgmh.org.tw; 5Department of Otolaryngology, Chang Gung Memorial Hospital, Chiayi Branch 613, Taiwan; yaote1215@gmail.com (Y.-T.T.); scm00031@gmail.com (C.-M.H.); 6Faculty of Medicine, College of Medicine, Chang Gung University, Taoyuan 333, Taiwan

**Keywords:** PTLD, polymorphism, lymphoma, nasopharynx, hematogenic disorder

## Abstract

(1) Background: Post-transplant lymphoproliferative disorder (PTLD) is a hematological disease and occurs because of immunosuppression after organ transplantation. Only a few studies have reported PTLD in the nasopharynx. In most cases, PTLD developed after solid organ transplantation, and cases of PTLD after bone marrow transplantation, are uncommon. (2) Case presentation: We report the case of a 40-year-old woman with myelodysplastic disorder who underwent hematopoietic stem cell transplantation (HSCT). After 3 months, she developed low-grade fever, progressive nasal obstruction, and bloody rhinorrhea. Endoscopy revealed a mass completely occupying the nasopharynx. A polymorphic PTLD was diagnosed on the basis of histopathological examination results. Reduction in immunosuppression and low-dose radiotherapy were prescribed for treatment. After a 3-year follow-up, no recurrence of PTLD or myelodysplastic disorder was detected. (3) Conclusions: While nasopharyngeal PTLD is rare, a routine examination of the nasopharynx should be considered in the post-transplant follow-up of patients for early detection and treatment of PTLD.

## 1. Introduction

Immunosuppressive agents used to reduce post-transplantation rejection reactions might cause serious complications, such as post-transplant lymphoproliferative disorder (PTLD), a hematological disorder strongly associated with Epstein–Barr virus (EBV) infection [1]. According to the fourth edition of the World Health Organization lymphoma classification, PTLD comprises four subgroups based on histopathological features, including early lesions (florid follicular hyperplasia, plasmacytic hyperplasia, and infectious mononucleosis-like lesions), polymorphic PTLD, monomorphic PTLD (B cell lymphomas and T/NK lymphomas), and classic Hodgkin lymphoma PTLD [2].

Only a few studies have reported PTLD of the nasopharynx, and most of the PTLDs were caused after solid organ transplantation [3]. Herein, we report a rare case of polymorphic PTLD with nasopharyngeal manifestation after hematopoietic stem cell transplantation (HSCT) and present the clinical images and histopathological features. In addition, we reviewed the literature associated with PTLD of the nasopharynx and compared the clinical manifestations, treatment, and prognosis with those manifested in our patient.

## 2. Case Presentation

A 40-year-old woman with myelodysplastic syndrome with excess blasts-I (MDS-EB-1; 2016 WHO MDS classification) (chromosome 46, XX (20)) underwent HSCT [3]. The immunosuppressant regimen was cyclosporine (400 mg/day) and prednisolone (20 mg/day). Then, three months after HSCT, she presented with intermittent low-grade fever, progressive nasal blockage, and bloody rhinorrhea. 

Nasopharyngoscopy revealed an exophytic mass completely occupying the nasopharynx (Figure 1). The surface of the tumor was necrotic and showed multiple bleeding spots. Computed tomography revealed an ill-defined tumor with heterogeneous enhancement at the nasopharynx (Figure 2). Blood biochemical examination showed that Epstein–Barr viral capsid antigen immunoglobulin G (EB-VCA IgG: 107.0 U/mL; the normal value is less than 20 U/mL) and Epstein–Barr nuclear antigen antibody (EBNA-Ab: 278.0 U/mL > 18 U/mL) display positive reactions, while EB-VCA IgM (10.0 U/mL < 36 U/mL) and EB early antigen antibody (EBEA-Ab: 5.0 U/mL < 9 U/mL) presented negative reactions. The quantitative Polymerase Chain Reaction (q-PCR) measurement of EBV DNA (in-house PCR assay; specimen: 5ml whole blood; target gene: BamHI-W gene, 3-kb repeat sequence contained in the EBV genome; detection limit: 200 copies/mL) showed a positive reaction (255 copies/mL). 

After biopsy was performed, hematoxylin–eosin staining revealed diffuse polymorphous infiltrates of lymphoid and plasma cells with scattered mitotic features in the nasopharyngeal mass (Figure 3A,B). Immunohistochemical staining disclosed that a majority of the lymphoid cells were positive for CD20 (Figure 4A), CD79A (Figure 4B), and multiple myeloma 1 (MUM1) (Figure 4C), partially positive for CD38 (Figure 4D) and positive for CD30 (Figure 4E). The lymphoid cells were negative for CD3, CD5, cyclin D1, CD10, Bcl-6, CD56, and CD23. In addition, Epstein–Barr encoding region (EBER) in situ hybridization showed positive results (Figure 4F), and immunostaining for immunoglobulin light chains (kappa and lambda) was negative. (S1 and S2)

After the patient was diagnosed as having polymorphic PTLD, we used local radiotherapy at a dose of 40 cGy for 20 fractions, and the immunosuppressant regimen was adjusted. At the 3-month follow-up, the nasopharyngeal lesion was in complete remission, and the following EBV DNA also dropped to the normal level (EBV DNA qPCR: 0 copies/mL). No recurrence of PTLD or MDS was detected during the 3-year long-term follow-up.

The patient was willing to provide the relevant medical records and image information for the publication and signed the consent for authorization. The publication of this case report was approved by the Institutional Review Board of Chang Gung Memorial Hospital (No.: 202001753B0).

## 3. Discussion

PTLD is a potentially fatal complication in recipients of organ transplantation and immunosuppressive medications. Reports of PTLD in the nasopharynx are quite rare, and most of them have presented cases in which PTLD developed after solid organ transplantation [3]. We present a rare case of nasopharyngeal PTLD after HSCT. 

Akbas et al published a large-scale PTLD study in 2015. A total of 276 PTLD patients were included in the article. The distribution sites included sinus, nasopharynx, oropharynx, larynx, esophagus, gastrointestinal tract, nodal, and extra-nodal regions. Among them, there were 17 nasopharynx PTLD identified (17/276: 6.16%), all of which were early lesions [4].

In Europe and the United States, over 80% of all PTLD cases are EBV-associated [5], indicating that EBV can be a major risk factor for PTLD. EBV infection is primarily transmitted through the oral route and manifests as infectious mononucleosis, which predominantly occurs in early childhood and lasts lifelong. Over 90% of the world’s adolescent and adult population are EBV carriers. In immune competent individuals, the cytotoxic T-cell response is a major factor in controlling expansion of EBV-infected cells [6]. Immunosuppression could lead to depressed T-cell function, which can cause consequent B-cell proliferation. Therefore, immunosuppression is a primary risk factor for PTLD. 

In 1995, the first case of PTLD in the nasopharynx was reported [1]. Since then, only a few studies have discussed the issue. We performed a review of such studies in the English literature and have listed those discussing nasopharyngeal PTLDs in Table 1 to compare the demographic characteristic, histopathological features, types of organ transplantation, clinical manifestations, therapeutic methods, time to PTLD of nasopharynx development after transplantation, and prognosis after treatment [1,7,8,9,10,11]. In general, compared with early lesions of the nasopharyngeal PTLD, polymorphic, monomorphic, and classic Hodgkin lymphoma PTLD have more aggressive courses and worse prognoses. Herein, we mainly focused on the comparison of the cases classified by the latter three.

Including our case, a total of nine studies discussing nasopharyngeal PTLD were selected after review (Table 1). They involved five male and four female patients, with their age ranging from 6 months to 40 years. In total, five of the eight PTLDs were polymorphic, and the other four were monomorphic. In the four monomorphic lesions, three were of B-cell type and one was of T-cell type. Overall, seven of the nine patients underwent solid organ transplantation, including four livers and three kidneys. The other two underwent bone marrow transplantation. On comparing histopathological features, we observed that the type of nasopharyngeal PTLDs in patients undergoing liver or bone marrow transplantation mostly belong to polymorphic type, whereas the type in patients with kidney transplantation was all monomorphic type. However, because the number of cases was scarce, it was insufficient to infer the correlation between the types of transplanted organs and the resulting PTLD based on this analysis. The time from organ transplantation to the formation of nasopharyngeal PTLD ranged from 3 to 61 months. Clinical signs and symptoms of those PTLD cases included fever, sore throat, progressive nasal obstruction, bloody rhinorrhea, pharyngitis, otitis media, enlarged tonsils and adenoids, and lymphadenopathy. 

The PTLD treatment in those cases included adjustment and tapering of immunosuppressive agents, antiviral drugs (acyclovir), chemotherapy (cyclophosphamide, adriamycin, vincristine, prednisone, and methotrexate), and anti-CD20 antibodies. Regarding prognoses, only one patient died (1/9, 11.1%). 

The primary goal of treating PTLD is to cure the disease, although it also carries the risk of graft rejection when immunosuppression is reduced, and preservation of allogeneic organs should also be considered [8]. Thus far, PTLD treatment is debatable. The reduction in immunosuppression has been widely documented in solid organ transplantations with the aim of restoring natural EBV-specific immune surveillance, and the response rate was reported as 20–50% [8]. In our case, we modulated immunosuppressive agents and added low-dose radiotherapy for enhancing the local therapeutic effect to obtain a favorable result. No recurrence of nasopharyngeal PTLD or MDS was detected during a 3-year follow-up.

## 4. Conclusions

Development of a nasopharyngeal PTLD following HSCT is a rare but a potentially fatal condition. Clinicians should pay attention to potential complications in patients undergoing organ transplantation, and a routine pharyngeal examination for those patients should be considered to facilitate immediate diagnosis and early treatment.

## Figures and Tables

**Figure 1 healthcare-09-00217-f001:**
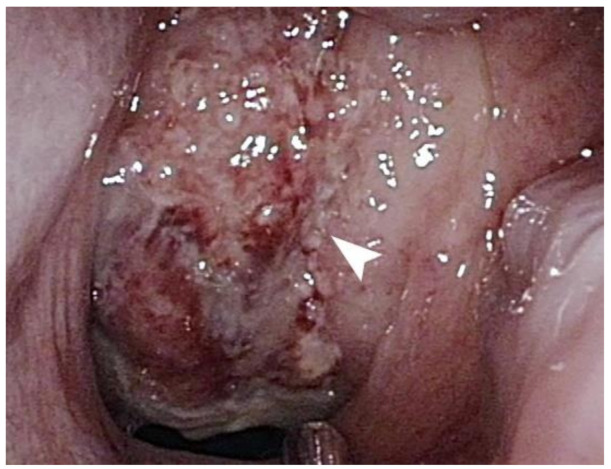
A mass, indicated by a white arrow, with a necrotic surface and multiple bleeding spots occupies the entire nasopharynx.

**Figure 2 healthcare-09-00217-f002:**
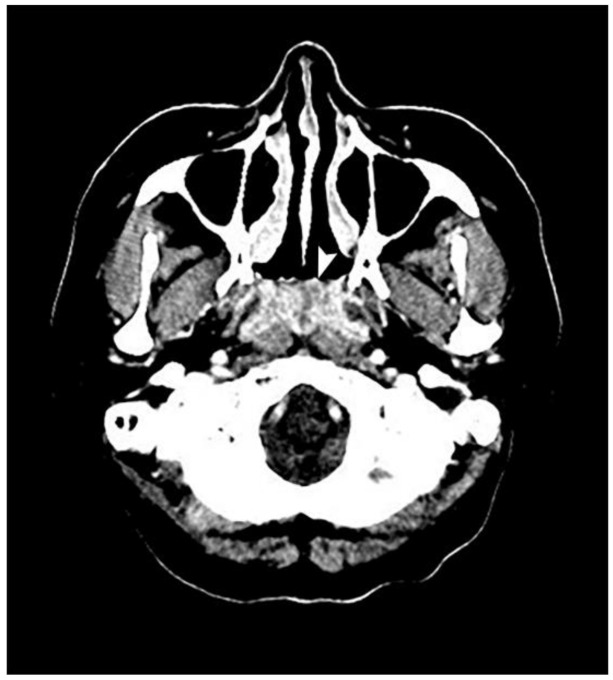
An axial view of a computed tomography scan reveals a mass, indicated by a white arrow, in the entire nasopharynx with ill-defined margins and heterogenous enhancement but that did not extend to the lateral pharyngeal space.

**Figure 3 healthcare-09-00217-f003:**
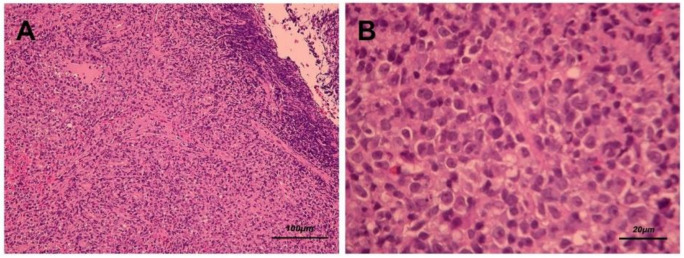
Hematoxylin and eosin staining (**A.** 100× and **B.** 400×) presents diffuse polymorphous infiltrates of lymphoid and plasma cells with scattered mitotic features in the nasopharyngeal mass.

**Figure 4 healthcare-09-00217-f004:**
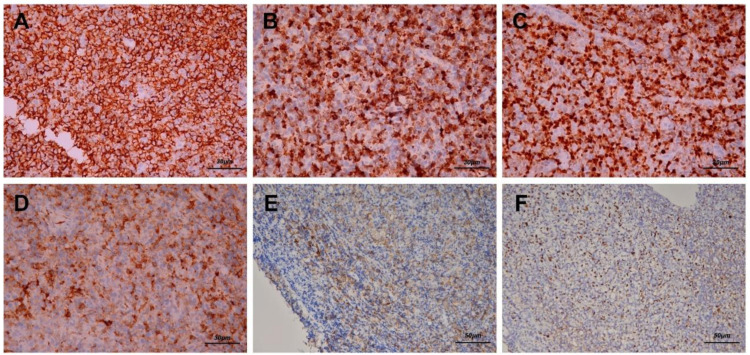
Immunohistochemical staining reveals that the majority of the lymphoid cells were positive for CD20 (**A,** 200×), CD79A (**B**, 200×) and multiple myeloma 1 (MUM1) (**C**, 200×). Some lymphoid cells were positive for CD38 (**D**, 200×) and positive for CD30 (**E**, 200×), which indicated that the infiltrates were B cells and plasma cells. In addition, Epstein–Barr encoding region (EBER) in situ hybridization showed positive results (**F**, 200×).

**Table 1 healthcare-09-00217-t001:** Summary of published studies on PTLD in the nasopharynx.

Study	Age	Sex	Types	Transplant	Time to PTLD	Treatment	Follow	Clinical Manifestations	IHC ^h^ stains
Lones MA (1995)	6m *	M	Polymorphous	Liver	26m	RI ^b^ and Acyclovir	19m	fever, pharyngitis, otitis media, enlarged tonsils and adenoids, diffuse LAPs ^g^	CD20, EBER ^i^
Lones MA (1995)	10m	M	Polymorphous	Liver	30m	RI and Acyclovir	28m	sinusitis, otitis media, enlarged adenoids, neck LAPs	CD20, EBER
Lones MA (1995)	35m	M	Polymorphous	Liver	61m	RI and Acyclovir	28m	enlarged tonsils and adenoids	CD20, EBER
Vargas H (2002)	38y ^#^	F	Monomorphic (B-cell)	Kidney	8m	RI and Acyclovir	12m	nasal obstruction, facial pain	CD20, CD46
Kfoury HK (2010)	49y	M	Monomorphic (T-cell)	Kidney	48m	RI, CHOP ^c^ and MTX ^d^	NR ^f^	sore throat for 1y	CD3, CD4, CD56
Spinato L (2011)	12y	F	Monomorphic (B-cell)	Kidney	9m	RI, Acyclovir and anti-CD20 antibodies	Death (1m after PTLD)	nasal obstruction, tonsillitis	CD20, CD79
Zhenyang Gu (2014)	26y	F	Polymorphic	HSCT ^a^	5m	RI alone	5y	foreign body sensation in pharynx	CD20, CD45, CD3, EBER
Zeren Baris (2018)	4y	M	Monomorphic (B-cell)	Liver	4m	Rituximab and chemotherapy	6.5y	treatment-resistant nasal stuffiness	CD20, EBER
Luan (2020) (Our study)	40y	F	Polymorphic	HSCT	3m	RI and RT ^e^	3y	low-grade fever, nasal obstruction, bloody rhinorrhea	CD20, CD79A, CD30, EBER, MUM1 ^j^

Abbreviation: * m, month; ^#^ y, years; ^a^ HSCT, hematopoietic stem cell transplantation; ^b^ RI, Reduced immunosuppression; ^c^ CHOP, Cyclophosphamide, Adriamycin, vincristine, and prednisone; ^d^ MTX, methotrexate; ^e^ RT, radiotherapy, ^f^ NR, no record; ^g^ LAP, lymphoadenopathy; ^h^ IHC, immunohistochemical stains; ^i^ EBER, EBV-encoded RNA; ^j^ MUM1, Multiple Myeloma 1.

## Data Availability

All data generated or analyzed during this study are included in this published article.

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
