# Peer review of "Malignant Post-Transplant Lymphoproliferative Disorder of Nasopharynx in Myelodysplastic Disorder"

_healthcare, 2021, doi:10.3390/healthcare9020217_

Round 1
Reviewer 1 Report
The case report by Luan et.al. is well-written and concise and presents a case of nasopharyngeal PTLD after haematopoetic stem cell transplantation. The brief litterature review is informative. I have a few minor comments:
- Information regarding EBV-status and EBER positivity is missing and needs to be added for the case report to be relevant.
- The evidence available in the litterature is to scarce to draw any conclusions, thus phrases such as ”kidney transplantation lead to monomorphic PTLD” need to be rephrased, this might just be a chance finding.
- It is stated that the Institutional Review Board of Chang Gung Memorial Hospital have approved the case report but the will of the patient itself is not provided and should be added.
Reviewer 2 Report
This case report presents a case of putative polymorphic PTLD following stem cell transplantation for MDS. Although nasopharynx is one of the less common locations for PTLD, there is little novel information, and the case has been studied insufficiently.
The diagnosis relies on the presence of the diffuse polymorphic infiltrate, which is not sufficiently shown in the current pictures. Furthermore, the most important studies concerning the presence of EBV and its latency type are lacking. In addition, studies for CD 30, Ig light chains etc. need to be done.
Specific points:
Abstract:
To subdivide PTLD into benign and malignant categories as done by the authors is not done in this way. Although the hyperplastic lesions may be considered benign and the monomorphic PTLDs are clearly malignant, PTLD are spectrum of lymphoproliferations with a range of aggressiveness. In addition, polymorphic PTLD is simply polymorphic PTLD, and not classified either benign or malignant.
PTLD after allogeneic stem cell transplant is not “extremely rare”
Case Presentation:
- Please use new WHO classification of MDS
- A surface is not “rotten” in medical terminology
Discussion:
By just referencing case reports of PTLD in the nasopharynx, the authors likely will miss cases located in the same site from larger studies. The larger studies PTLD will have to be reviewed to judge how frequently the nasopharynx is site of the disease.
Reviewer 3 Report
The paper by Luan C-W et al presents a case of nasopharyngeal post-transplant lymphoproliferative disease (PTLD) in a 40-year-old female patient who underwent hematopoietic stem cell transplantation (HSCT) as treatment for myelodysplastic syndrome. There are only a handful of cases of nasopharyngeal PTLD described in the scientific literature, which is why the report by Luan et al is of significance.
Major concerns:
- The idea that most PTLDs are benign (as written on line 18 of the Abstract) is not supported by the literature. Other inconsistencies include the sentence in the Discussion that 55-65% of PTLDs are EBV-related. The actual figure is much higher. The authors must do a proper literature search on PTLD and provide appropriate references.
- The language used throughout the manuscript is often imprecise. For example, it is sometimes difficult to know when the authors are referring to PTLD as a general entity, and when they are specifically referring to nasopharyngeal PTLD. Some passages are difficult to comprehend. The second paragraph of the Discussion section is particularly confusing. I suggest that the authors consult with an expert on EBV infection, latency and PTLD.
Minor comments:
- Line 55 and line 70: replace the word “rotten” with “necrotic”.
- Line 58 and line 79 (legend to Figure 3): replace the word “diffusely” with “diffuse”. This sentence is confusing. I suggest rewriting it as “….containing lymphoid and plasma cells with scattered mitotic features in the nasopharyngeal mass (Figure 3A and 3B).”
- Figure 3 should include a size bar for each figure in the composite.
- Figure 4 should include a size bar for each figure in the composite.
- Figure 4, lines 85-86, contains negative data. This does not belong in the figure legend. Please remove from figure legend and incorporate it in the text.
- Discussion section line 95: replace “kissing disease” by “infectious mononucleosis”.
Round 2
Reviewer 2 Report
none, see my last review
Reviewer 3 Report
Revision 2:
The resubmitted paper is improved. However, I still have concerns which require correction. These concerns are mostly minor and are enumerated below:
- Abstract, line 21: remove the word “regarding”.
- Abstract, lines 29-32 (last sentence): please consider rewriting as follows: “While nasopharyngeal PTLD is rare, a routine examination of the nasopharynx should be considered in the post-transplant follow-up of patients for early detection and treatment of PTLD.”
- Case presentation, lines 61-63: The authors have added a statement that the qPCR test was above the threshold of 200 copies/mL. I assume that the test was performed on whole blood. Please specify specimen type in the text.
- Case presentation, line 71: Replace “chain” by “chains”, and remove the word “both”.
- Discussion, line 107: Replace “…and those nasopharynx PTLD were all early lesions.4 ”, by “…, all of which were early lesions.4 “
- Discussion, line 108: Replace “…of general PTLD cases is EBV-associated5 “, by “…of all PTLD cases are EBV-associated5 “.
- Discussion, line 109: Remove the word “general”. This word is not needed because the prior sentence uses the word “all”.
- Discussion, lines 111-112: “Over 90% of the world’s adolescent and adult population are EBV carriers, and the adenotonsillar region is the predominantly latent area.3” The second part of this sentence is problematic, primarily because it is not well-referenced and because it is too cryptic to contribute anything useful to this paper. For these reasons I suggest that the authors delete this phrase, and keep only the first part of the sentence.
- Discussion, lines 112-114: “In an immune competent population, the immune response of cytotoxic T cells could stabilize EBV infection and form latently infected memory B cells, resulting in the population remaining asymptomatic for life.5” While technically the information is more or less correct, the English quality is poor. I suggest that the authors simplify the statement into a shorter sentence by rewriting it as follows: “In immune competent individuals, the cytotoxic T-cell response is a major factor in controlling expansion of EBV-infected cells”. This is the work of Alan Rickinson, and should be referenced appropriately.
- Discussion, lines 116-119: The authors state, “Patients undergoing organ transplantation require immunosuppression to prevent rejection reactions. Herein, the latent EBV might become activated and induce an abnormal proliferation of B cells, which would contribute to the development of PTLD. In the presented case, based on the EBV-related serology, it indicated a latent infection of EBV.” First of all, not only organ transplant patients require immunosuppression; HSCT patients often receive ATG which is a potent immunosuppressive against T cells to combat GvHD. Secondly, the authors establish that the infection is latent by simply assessing the EBV serological markers. The problem is that the EBV serological response in severely immunosuppressed patients is unreliable. For these reasons, my recommendation is to consolidate these 3 sentences by simply replacing them with 1 sentence such as: “Therefore, immunosuppression is a primary risk factor for PTLD.”
- Discussion, line 120: Please remove the words “after transplantation”. These words are redundant because ‘PTLD’ means ‘post-transplant lymphoproliferative disease’.
- Discussion, line 147 and line 149: Please remove the word “general”. In this context it is understood that the authors mean all PTLD cases.
- Table 1, second line: Replace “Livr” by “Liver”.
Author Response
Please see the attachment.

This manuscript is a resubmission of an earlier submission. The following is a list of the peer review reports and author responses from that submission.